# Effect of SiO$_2$ and TiO$_2$ Nanoparticles on the Performance of UV Visible Fluorescent Coatings

**Sanja Mahović Poljaček** [1,*] , **Tamara Tomašegović** [1,*] , **Mirjam Leskovšek** [2] and **Urška Stanković Elesini** [2]

1. Faculty of Graphic Arts, University of Zagreb, Getaldićeva 2, 10000 Zagreb, Croatia
2. Faculty of Natural Sciences and Engineering, University of Ljubljana, Aškerčeva Cesta 12, SI-1000 Ljubljana, Slovenia; mirjam.leskovsek@ntf.uni-lj.si (M.L.); urska.stankovic@ntf.uni-lj.si (U.S.E.)
* Correspondence: sanja.mahovic.poljacek@grf.unizg.hr (S.M.P.); ttomaseg@grf.hr (T.T.)

**Abstract:** In the present research, the properties of ultraviolet (UV) visible (daylight invisible) fluorescent coatings modified by the addition of SiO$_2$ and TiO$_2$ nanoparticles were studied. Structural, surface, and mechanical properties and changes in the coatings caused by accelerated ageing were analyzed. The results showed that the addition of nanoparticles caused the changes in unaged and aged printed coatings. Reflectance measurements of modified coatings showed that addition of TiO$_2$ nanoparticles improved the visual effect of the unaged coatings. Furthermore, results have shown that the addition of SiO$_2$ did not diminish the reflectance of the modified coatings after ageing. The results of roughness measurements showed that the addition of SiO$_2$ decreased roughness after the ageing process, probably due to the degradation process indicated by Attenuated Total Reflection Fourier Transform Infrared (ATR-FTIR) spectroscopy. The roughness of the coatings with TiO$_2$ nanoparticles was increased after the ageing on the samples with higher concentrations of TiO$_2$ due to the agglomerates of plastisol formed on the surface of the coatings, visible in SEM images. Surface analysis of coatings showed that TiO$_2$ caused an increase in the polarity of the surface coatings. Results of the bending stiffness showed that the addition of the nanoparticles to the coating, especially of SiO$_2$, significantly improved the bending stiffness of the unaged samples.

**Keywords:** UV visible fluorescent coatings; nanoparticles; SiO$_2$; TiO$_2$

## 1. Introduction

UV visible (daylight invisible) fluorescent coatings (FC) belong to the group of luminescent materials that have the possibility of absorbing UV radiation and re-emitting of photons of a different radiation [1–3]. Fluorescence occurs immediately after light absorption and lasts only as long as the primary radiation acts, after which it stops almost immediately; it can be seen in the $10^{-4}$ to $10^{-9}$ s range [4]. FC are widely used in different industries. They can be used in decorative and packaging industries for different control markings, signaling, and orientation purposes, they can be used as functional coatings, coatings on safety devices, as UV sensors, in fluorescent lamps, for CRT display tubes, and for many other functions [5–8]. They can be used as special protective elements of printed products, i.e., document security applications, tagging of postage stamps, etc. [9–14].

Considering the fact that UV visible (daylight invisible) fluorescent coatings can be used for different applications and printed on different substrates, research on their properties is interesting from different aspects. Development of suitable material that meets the specific printing requirements and ensures the optimal functional properties is of high importance due to the wide application of these kinds of coatings. With this in mind, in this research, the properties of modified UV visible FC were studied. FCs were modified by addition of nanoparticles, bearing in mind that they can, to a certain extent, affect the structural, surface, and mechanical properties of the FCs and the visual response of the fluorescent phenomenon as well.

Studies published previously have researched the advantages and disadvantages of coatings containing nanomaterials. One of the most interesting nanomaterials for the paint industry are nanoscale titanium dioxide ($TiO_2$) and silicon dioxide ($SiO_2$). The potential benefits for coatings produced by addition of nanomaterials are different and miscellaneous. Some of them give the coating better antibacterial, fire-retardant properties, and scratch resistance, they can ensure better UV-protection and water-repellent properties, as well as anti-graffiti properties, anticorrosive properties, and other functionalities [15–21]. $TiO_2$ nanoparticles are extremely interesting in coating applications because of their photocatalytic activity and UV-protection. Nguyen et al. [22] researched the effect of rutile titania nanoparticles on the mechanical property, thermal stability, weathering resistance, and antibacterial properties of styrene acrylic polyurethane coating. Published results have shown that the addition of rutile $TiO_2$ nanoparticles improved the impact strength and adhesion of the coating. They proved that accelerated weathering tests indicated that rutile $TiO_2$ nanoparticles mitigated the chemical change, weight loss and mechanical degradation of the coating. Rommens et al. [23] in their research concluded that $TiO_2$ has a positive effect on the weather resistance of a coating and that it provides significant protection against UV radiation and degradation to the underlying resin molecules. Al-Kattan et al. [24] studied the release of titania from paints containing pigment-$TiO_2$ and nano-$TiO_2$ into the environment during weathering with water. They concluded that paints containing nano-$TiO_2$ released limited amounts of materials into the environment and suggested design options for hazard reduction. Kafizas et al. [25] studied titanium and composite metal/metal oxide titanium thin films on glass and photocatalytic activity of coatings. They concluded that all films, after certain exposure to irradiation, demonstrated super hydrophilicity (PSH) and that Ag:Au $TiO_2$ composite coating was found to be a useful coating due to its robustness, self-cleaning, and reuse properties. Solano et al. [20] synthesized $TiO_2$ and ZnO nanoparticles and evaluated the influence of those nanoparticles the anticorrosive ability, antibacterial ability, and self-cleaning efficiency of paints. They concluded that modification of paints with a low concentration of $TiO_2$ nanoparticles offers an improvement in the physicochemical properties of paints.

Colloidal silicon dioxide (fumed silica, $SiO_2$) has many applications in the paints and coatings industry. It is used as an anti-settling or suspension agent for fillers and pigments and it can be used as an electrically conductive coating containing metallic powders or lakes. It is also used for rheology control of paints according to the application to improve the gloss and coloristic properties and for scratch resistance of different coating systems [26–28]. Al-Kattan et al. [24] published research in which the characterization of the materials released from paint containing nano-$SiO_2$ during weathering and exposure to water was performed. On the other hand, Sung et al. [29] published a study that demonstrated the amount of nanosilica released into environment due to weathering, i.e., UV exposure. Jacobs et al. [30] researched surface degradation and nanoparticle release of a commercial nanosilica/polyurethane coating under UV exposure. Results demonstrated the degradation of the polyurethane matrix and accumulation of silica nanoparticles on the coating surface, and their release into the environment by simulated rain. Mizutani et al. [31] prepared a wall paint by utilizing a nano-composite emulsion that contained silica nanoparticles and polyacrylate. They concluded that the produced paint showed significant solvent resistance, excellent antipollution properties, and high flame resistance. Zhou et al. [32] published research of coatings embedded with different types of silica and concluded that coatings containing silica had improved abrasion and scratch resistance.

The presented published studies related to production of coatings for different applications showed that addition of nanomaterials into the base material can improve and enhance different properties of prepared coatings. Specifically, nanoscale $TiO_2$ and $SiO_2$ are widely used in the paint industry and to the best our knowledge a study on their addition to UV visible fluorescent coatings has not been published. In order to evaluate the properties of modified coatings, the changes caused by the ageing process, surface,

structural, and mechanical properties of printed coatings, and their visual responses were observed.

## 2. Materials and Methods

### 2.1. Preparation of the Samples

Nanocomposite—FCs consisted of the plastisol-based transparent base (TB) PLASTO-LAK K73990K1 by Epta Inks (Eptanova S.R.L., Milano, Italy), UV visible (daylight invisible) fluorescent red pigment (FP red) by Cestisa (added in the mass concentration of 3% to the TB) and varied amount of $SiO_2$ and $TiO_2$ nanoparticles. The mass concentration of $SiO_2$ nanoparticles added to the TB with FP red was set to 1%, 2%, and 3%, and the mass concentration of $TiO_2$ nanoparticles was set to 0.5%, 1%, and 1.5%. The portion of the $TiO_2$ nanoparticles in the coating was lower than that of $SiO_2$ because $TiO_2$ in higher concentrations can diminish the optical and visual properties of paints and coatings [20]. In this way, six variations of coatings were prepared, with the aim of analyzing the influence of the nanoparticles on the important properties of the FC. Properties of the used nanoparticles are presented in Table 1.

**Table 1.** Basic properties of $SiO_2$ and $TiO_2$ nanoparticles.

| Nanoparticles | Production Name | CAS No. | Average Primary Particle Size [nm] | Weight [%] |
|---|---|---|---|---|
| $SiO_2$ | Aerosil 200 | 112945-52-5 | 12 | >99.8 |
| $TiO_2$ | Titanium (IV) oxide, anatase | 1317-70-0 | 15 | 99.7 |

Modified coatings were prepared by mechanically mixing the TB and FP red and adding the defined amount of nanoparticles. The mixture was then homogenized using an ultrasonic disperser (UP100H Hielscher, Hielscher Ultrasonics GmbH, Teltow, Germany) for 2 min at 100% device amplitude.

Prepared coatings were screen printed on coated white UPM Digi Finesse (Augsburg, Finland) premium silk paper, using the screen printing plate with a mesh density of 40 lines $cm^{-1}$. The paper had grammage of 300 $g \cdot m^{-2}$ and was conditioned prior to the printing at a temperature of $24 \pm 1\,°C$ and 50–55 % relative humidity. Paper chosen for this research is a commonly used printing substrate that can be applied to the reproduction of various graphic products. The printing plate's non printing areas were produced using AZOCOL Z 133 emulsion by KIWO (Wiesloch, Germany). Expos-it VASTEX unit, model E2331 (Vastex International, Inc., Bethlehem, PA, USA) was used for the purposes of exposure, and drying of the printing plate was conducted in Dri-Vault screen drying cabine (Vastex International, Inc., Bethlehem, PA, USA). Developing of the printing plate was performed using water. The printing process was performed using a screen printing machine and the printed samples were air-dried for 48 h at a temperature of $25 \pm 2\,°C$.

### 2.2. Ageing Process

The ageing process was performed with the aim of analyzing the resistance of the UV visible fluorescent coating to photo-degradation; and to determine if the added nanoparticles can improve the photostability of the prepared coatings, alongside other important coating properties. Papers published in the past have shown that the addition of components like nanoparticles into the base material, may either have a protective effect on or accelerate the photo-degradation of the coating [33,34].

In this research, the laboratory ageing process was performed by exposure of the samples to xenon radiation in a test chamber Solarbox 1500e (CO.FO.ME.GRA., Milano, Italy). An outdoor filter, soda lime glass UV, was used to simulate the outdoor exposure to daylight. Irradiation was set to 550 $W \cdot m^{-2}$ with the 50 °C temperature. The equipment was set in accordance with the ISO 4892-2 standard [35]. All samples were subject to artificial ageing for 6 and 12 h. It is important to emphasize that the shortness of the ageing duration was necessary due to the generally poor lightfastness of FPs [36], and because the ageing

process served primarily to assess the coating's lifespan, since the UV radiation is necessary for the fluorescent effect.

### 2.3. Characterization Methods

Spectral reflectance of the printed coatings was measured using the Ocean Optics USB 2000+ spectrometer (Ocean Optics, Orlando, FL, USA) and Deuterium-Tungsten Halogen UV light source DH-2000. In this way, the reflectance of the FP could be observed in the area between 525 and 625 nm. Due to the UV light source, optical brighteners in the paper caused the total reflectance of >160% at some wavelengths [37,38], but that did not interfere with the area of emission of the FP.

ATR-FTIR spectroscopy was used to identify the presence of functional groups of interest in the surface of the FCs applied onto paper. In this research, ATR-FTIR was specifically used to characterize the changes that occur in the coating components (TB and FP with the addition of nanoparticles) after the process of artificial ageing. ATR-FTIR analysis was performed using Shimadzu IRAffinity-1 FTIR Spectrophotometer (Shimadzu Corporation, Kyōto, Japan) with 15 scans per sample.

Surface roughness can significantly affect the functional properties of the coatings, especially in cases when special effect coatings are observed. In this paper, the surface roughness on printed coatings was measured due to the fact that the visual effect of fluorescence may correlate with the surface roughness, according to Zheng et al. [39]. The choice of profiling methods and roughness parameters is defined by international standards (ISO 11562, DIN 4777, DIN 4762) [40]. For the purpose of this paper, a basic roughness parameter was measured, $R_a$—the arithmetic mean deviation of the profile (DIN 4768, ISO/DIS 4287-1) [41]. The instrument used to measure roughness was MarSurf PS 10 (Mahr GmbH, Göttingen, Germany) with the stylus method. The size of the stylus was 2 μm and the measuring force was 0.00075 N. The average of ten surface roughness measurements was taken for each sample.

Contact angles of water on the printed coatings were measured using the Data Physics OCA 30 goniometer (DataPhysics Instruments GmbH, Filderstadt, Germany). Contact angles were measured using the sessile drop method eight times on each sample at different positions. The shape of the drop was a spherical cap, and the volume of the drop was 1 μL. All measurements of the contact angles were performed at 2 s after the drop had touched the coating surface. Measurement of the contact angles of water was necessary to obtain the information on the changes of the surface polarity after the addition of different nanoparticles to the coating and after the artificial ageing.

Bending stiffness of the papers printed with FCs was measured using the Lorentzen & Wettre bending tester (ABB AB/Lorentzen & Wettre, Zurich, Switzerland). The L&W Bending Tester is used to determine the stiffness of paper, cardboard, or board according to the bending or the resistance that the material provides at the selected bending angle. The bending stiffness of the samples was measured in order to analyze the influence of the nanoparticles and of the ageing process on the mechanical properties of the modified coatings. Bending stiffness was measured three times for each type of printed sample at the angle of 7.5°.

Visual evaluation of the coating surfaces before and after the ageing process was carried out using the Olympus BX51 microscope (Tokyo, Japan) at a magnification of 20×. Images of the unaged and aged FC surfaces were taken under UV light source to observe the visual changes in fluorescence after the ageing process.

Surfaces of the coating samples were observed by SEM microscope SEM JSM-6060 LV (Jeol, Tokyo, Japan). The instrument is designed for surface, morphology and topography studies, as well as for the determination of the particle size. To assure the uniform electrical properties of the samples, samples of the UV fluorescent printed layers were coated with an ultra-thin layer of gold after which they were fixed on a specimen stub. Images were captured at different magnifications. SEM images have proven to be helpful for the

understanding of the surface morphology of printed nanocomposite coatings and in the analysis of the changes in the surface properties that occur after the artificial ageing process.

## 3. Results and Discussion

### 3.1. Morphology of the Input Materials

For observation, the viscous TB was applied in one layer to a non-stick backing paper and allowed to air dry for 48 h. The layers were then peeled off a paper, attached onto the specimen holders, and covered with an ultra-thin layer of gold (with high-vacuum evaporation). As can be seen in Figure 1, spheres up to 5 μm in diameter are present in the sample. Since the suspension of the TB is plastisol, the observed small spheres in Figure 1 could be particles of a polymer resin dispersed in an aqueous medium–plasticizer [42]. Although plastisol undergoes gelation and "fusion" under the influence of temperature or sol–gel transition (physical gelation) [43], the small spheres of polymer could still be present in the plastisol layer.

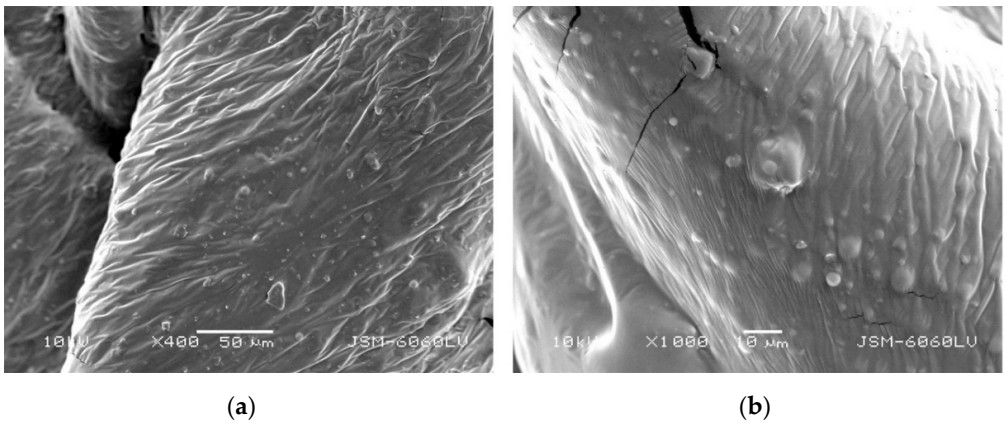

| (a) | (b) |

**Figure 1.** Transparent base, mag. (**a**) 400× and (**b**) 1000×.

The surface structure and cross section of TB printed in screen printing on coated paper is presented in Figure 2. One can see that the surface is relatively rough with micro voids and polymer spheres.

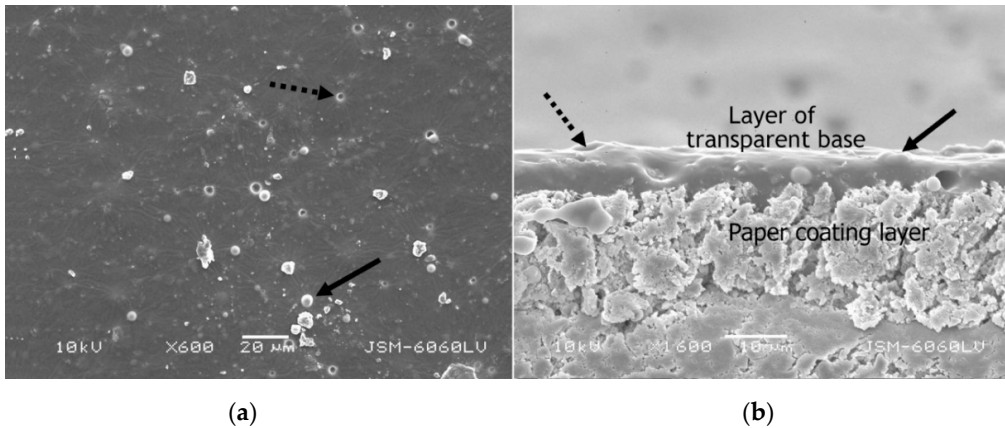

| (a) | (b) |

**Figure 2.** (**a**) Surface morphology (mag. 600×); (**b**) cross-section of transparent base applied on coated paper and (mag. 1600×); solid arrow points to polymer sphere, dotted arrow points to micro void.

Figure 3a shows particles of different size (from a few microns up to 80–100 μm) and shape of the FP. One can see the spatial structure of the pigment's flaky and crispy nature with loosely aggregated particles. Figure 3b shows nanoparticles of $SiO_2$. According to the technical specification, the individual particles of silica are cca. 12 nm in size. However, larger, micrometer-sized agglomerates can be seen in Figure 3b. As stated by Patel

et al. [44], aggregates are fused from primarily particles, stabilized by hydrogen bonding and electrostatic interactions, and have a large specific surface area and hydrophilicity.

Figure 3c presents the nanoparticles of $TiO_2$. According to the technical specification, the individual particles of $TiO_2$ are up to 15 nm in size. Figure 3c shows micron-sized agglomerates of $TiO_2$ nanoparticles. These agglomerates are usually formed during the drying stage of nanosuspensions to nanopowders and are in a relatively stable form (usually hardly segregated), which reduces the surface area to volume ratio of nanoparticles and thus affects their effective application.

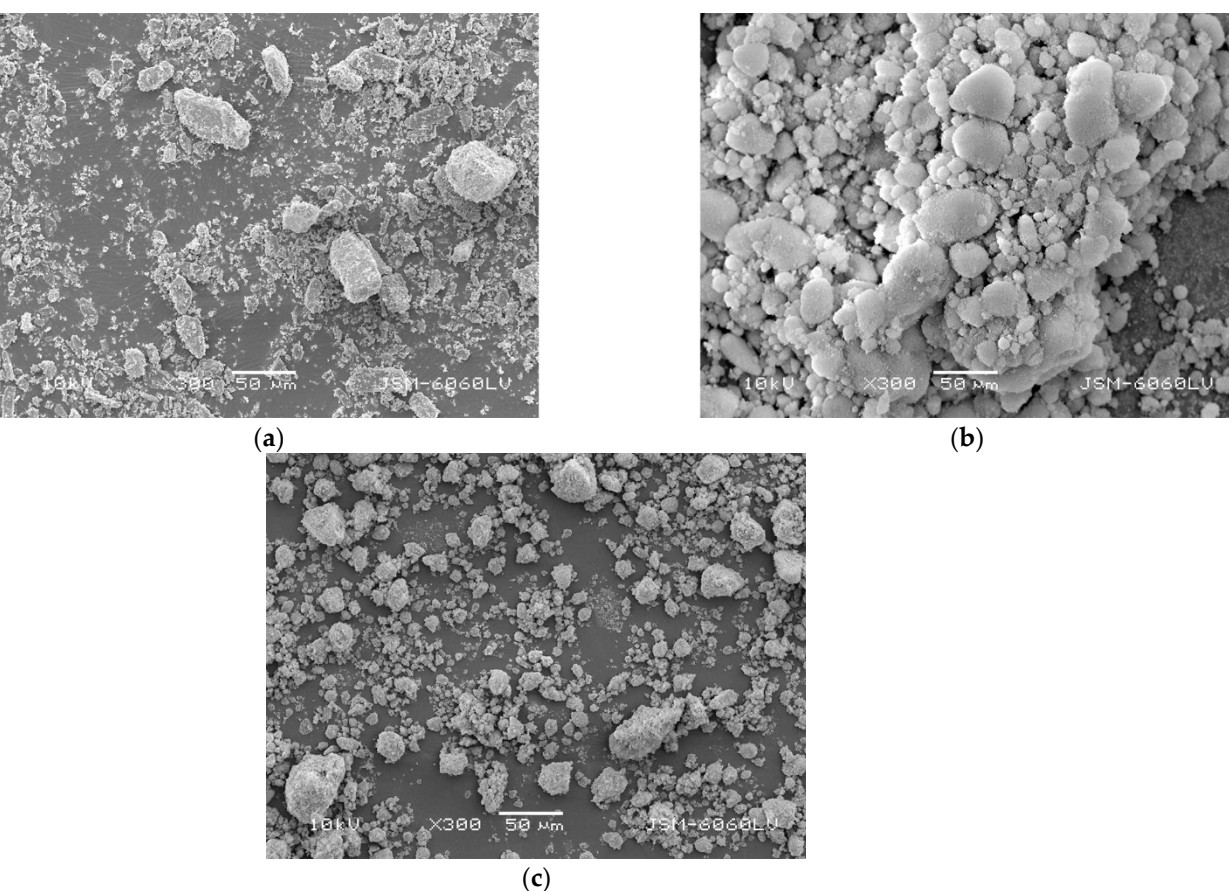

**Figure 3.** (**a**) UV visible (daylight invisible) fluorescent red pigment as received; (**b**) silicon dioxide (fumed silica) as received; (**c**) titanium (IV) oxide as received (mag. 300×).

### 3.2. Spectral Reflectance of FCs

The results of spectral reflectance measurements of the prepared coatings are presented in Figures 4–6. In Figure 4a spectral reflectance of the TB without the FP and without the nanoparticles can be seen. The reflectance over 100% in the area between 420 and 470 nm can be assigned to the optical brighteners in the paper [38]. After the ageing of 6 and 12 h, the reflectance in this area was decreased, and the reflectance in the area between 350 and 400 nm was increased. These changes can be assigned to the changes in the paper due to the process of artificial ageing, since no changes in the surface structure of the TB were visible in ATR-FTIR spectra of the TB (Section 3.3). In Figure 4b one can see the spectral reflectance of the unaged and aged coating containing the 3% of FP mixed in the TB (without the nanoparticles). Peaks detected in the range of 580–630 nm correspond to the spectral reflectance of the used FP. It is visible that the aging process caused a decrease in the spectral reflectance through the visible range spectrum. Obviously, the degradation process in the coating occurred, causing the reduction in the visual effect of the fluorescence.

Figure 5 presents the reflectance spectra of the FCs with the addition of 1%, 2%, and 3% of SiO₂, respectively. One can see that the addition of nanoparticles does not change the spectral reflectance of coatings; the presented reflectance curves of unaged samples with addition of SiO₂ are the same as the spectral reflectance of coatings without nanoparticles (Figure 4b).

It can be observed that the effect of the optical brighteners in the paper is not as pronounced as for the coating consisting solely of TB, since both FP and nanoparticles have decreased the transparency of the prepared coatings. However, the ageing process has a significant impact on the spectral reflectance of the FC, especially after 12 h. One can see that the spectral reflectance is decreased causing the fading of the visual effect of fluorescence.

In Figure 5 one can see that the concentration of the SiO₂ nanoparticles in the coating plays a significant role in the ageing process. It is visible that the addition of lower concentrations of SiO₂ (1%) resulted in the smallest decrease in the FC's reflectance after 12 h of ageing (Figure 5a). This is in accordance with recent research, where SiO₂ nanoparticles proved to have a better impact on the suppression of ageing of the nanocomposite when added in smaller amounts [45].

Figure 6 presents the reflectance spectra of the FCs with the addition of 0.5%, 1%, and 1.5% of TiO₂, respectively. One can see that the addition of nanoparticles of TiO₂ caused a slight increase in spectral reflectance % of fluorescent coating in comparison to spectral reflectance of coating without nanoparticles (Figure 4b). The changes in the concentration of TiO₂ nanoparticles did not play any significant role in the reflectance of the FC, but the coating samples with TiO₂ nanoparticles did display visually more saturated fluorescence that was visible in the highest spectral reflectance compared to the samples with SiO₂ nanoparticles.

Since SiO₂ nanoparticles possess UV-reflective properties [46,47], they have proven more effective at ageing protection of FC compared to TiO₂ nanoparticles (Figures 5 and 6). Keeping in mind that TiO₂ nanoparticles have photocatalytic properties [48,49], they could have initiated the UV-induced changes in the prepared coatings that were visible not only for the decreased reflectance of the FC after ageing for 6 and 12 h, but in the general changes of the surface polarity presented in Section 3.5 via measurements of the water contact angle.

The measurements of spectral reflectance of FC with the addition of SiO₂ or TiO₂ nanoparticles presented peaks in the range of 580–630 (Figures 4b, 5 and 6). They can be assigned to Eu³⁺–based FP [50].

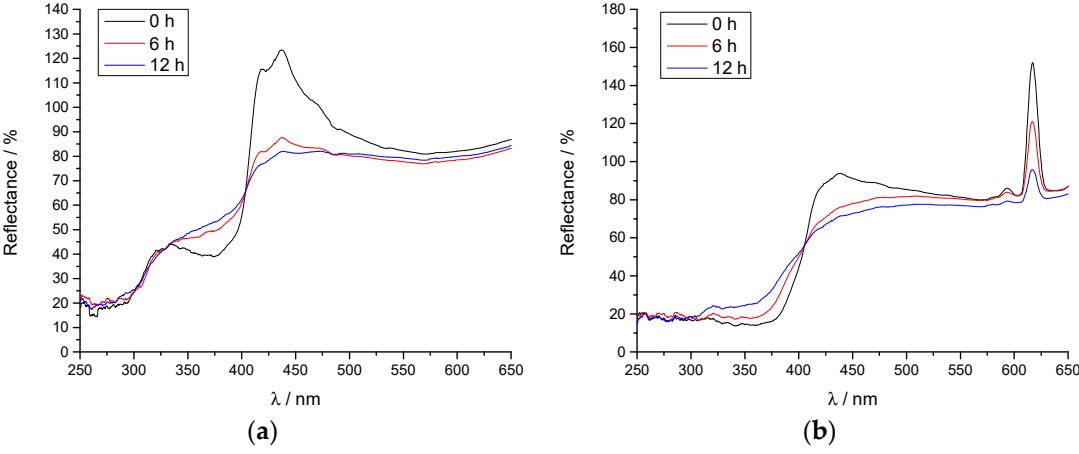

**Figure 4.** Spectral reflectance of unaged and aged samples: (**a**) transparent base; (**b**) TB + FP (without the nanoparticles).

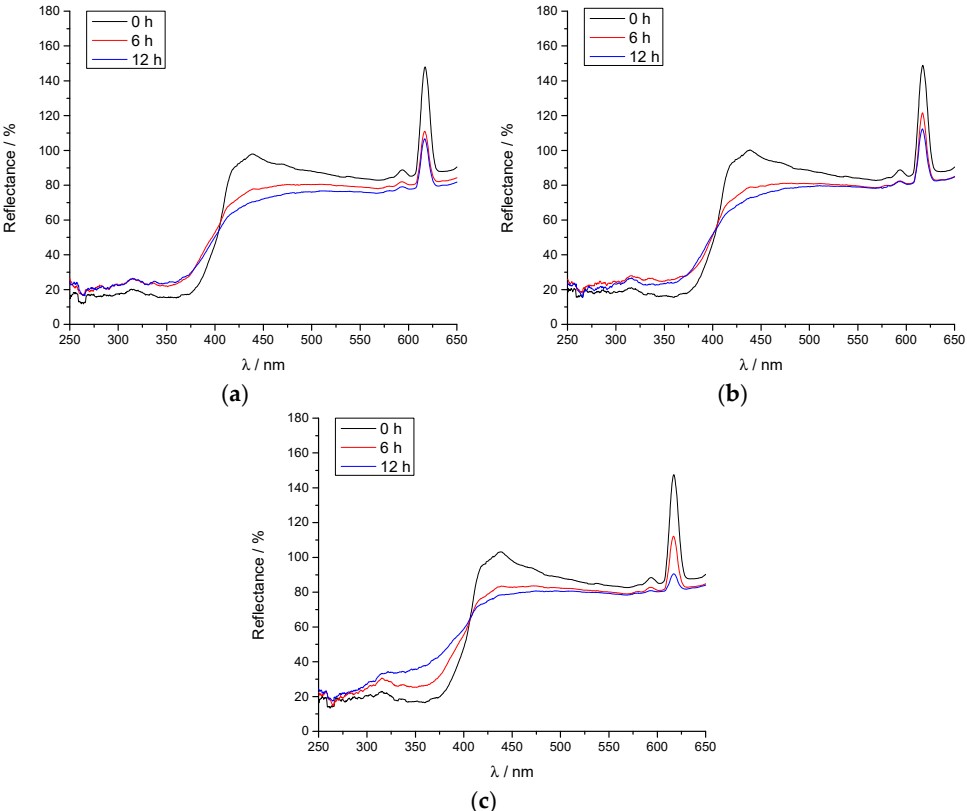

**Figure 5.** Spectral reflectance of unaged and aged coatings: (**a**) FC + 1% SiO$_2$; (**b**) FC + 2% SiO$_2$; (**c**) FC + 3% SiO$_2$.

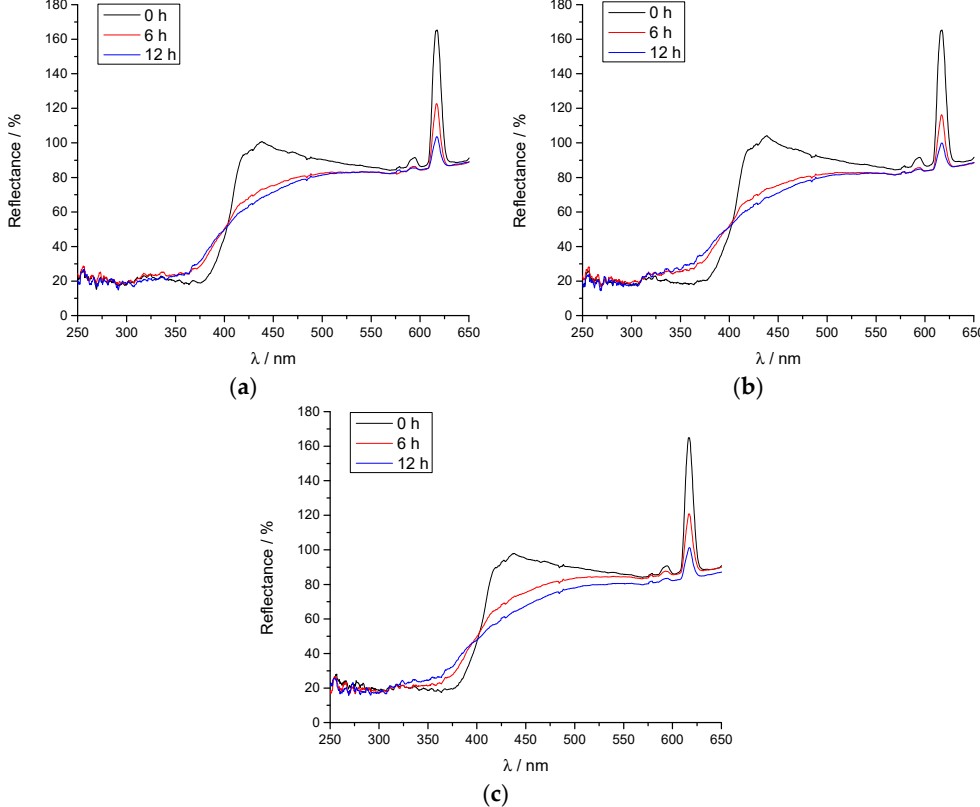

**Figure 6.** Spectral reflectance of unaged and aged coatings: (**a**) FC + 0.5% TiO$_2$; (**b**) FC + 1% TiO$_2$; (**c**) FC + 1.5% TiO$_2$.

### 3.3. ATR-FTIR Spectra of FCs

Results of the ATR-FTIR analysis of unaged and aged samples are presented in Figure 7. The main aim of the analysis was to detect the changes in the functional groups in the coatings' surface, which could point to the degradation due to the ageing process and to the changes in the surface properties, i.e., polarity of the surface.

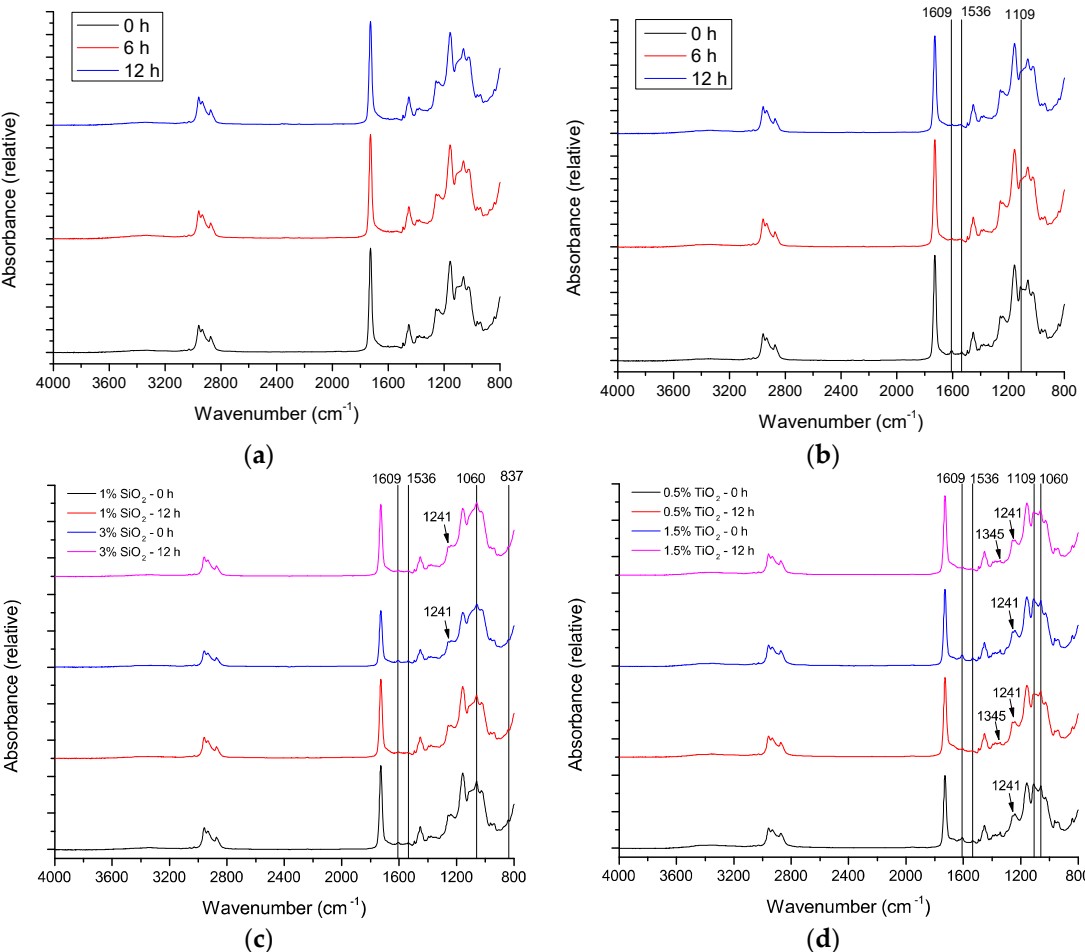

**Figure 7.** FTIR-ATR spectra of unaged and aged (**a**) transparent base; (**b**) FC without the nanoparticles; (**c**) FC with added SiO$_2$ nanoparticles; (**d**) FC with added TiO$_2$ nanoparticles.

In Figure 7a, which presents the ATR-FTIR spectra of TB without the FP and nanoparticles, one can see that the ageing process did not have any effect on the TB that is visible on ATR-FTIR spectra. It can be concluded that the plastisol base is stable after 12 h of artificial ageing. All effects of the ageing process manifested only when FP and nanoparticles were added to the base. Figure 7b presents the ATR-FTIR spectra of the TB with added FP. The changes after the ageing process are visible at 1109, 1536, and 1609 cm$^{-1}$ and will be discussed in the analysis of the samples with both FP and nanoparticles. These peaks correspond to the plane bending vibration of C–H bond, a –CONH– absorption band related to the Eu$^{3+}$ based UV luminescent pigment, and C=O stretching vibrations, respectively.

When comparing Figure 7c,d, it is visible that the addition of SiO$_2$ nanoparticles caused slightly different changes in ATR-FTIR spectra than the addition of TiO$_2$ nanoparticles. Specifically, the peak at 837 cm$^{-1}$ in Figure 7c can be assigned to C–Cl bond plastisol [51]. The absorbance of this peak was decreased for the coating with 3% SiO$_2$, after 12 h of artificial ageing, pointing to the possible start of the degradation of the TB [52]. The second significant change was present in the area of 1060 cm$^{-1}$, with the increased absorbance of the peak for the coating with 3% of SiO$_2$, which could be related to the C–O–C stretching

in plastisol [53]. This peak is of significantly lower absorbance for the coating samples with $TiO_2$ (Figure 7d). The peak at 1241 $cm^{-1}$ for the samples with $SiO_2$ nanoparticles presented with the decreased absorbance for both unaged and aged coatings compared to the coatings with $TiO_2$ nanoparticles. It can be assigned to C–O stretching [54]. It is important to notice that the samples with $TiO_2$ nanoparticles have higher absorbance in this area, which is visible in other polar groups in ATR-FTIR analysis, as well, and is in accordance with higher polarity of the coating surfaces with $TiO_2$. Peaks at 1536 and 1609 $cm^{-1}$ are common for the coatings with both types of nanoparticles in this research. The only difference is that their absorbance was higher for the unaged coatings with $TiO_2$ nanoparticles. However, their absorbance visibly decreased after the ageing process for the coatings with $TiO_2$ nanoparticles. Their absorbance was not as significantly decreased for the coatings with $SiO_2$ after the artificial ageing—but for these coatings, they were not initially as pronounced. The peak at 1536 $cm^{-1}$ corresponds to the –CONH– absorption band, related to the $Eu^{3+}$ based UV luminescent pigment [55]. The decreased absorbance of this peak was present after the ageing process for all samples, but it is more pronounced for the samples with $TiO_2$. This result is in accordance with the results of the spectral reflectance measurements, where the coatings with $TiO_2$ displayed decreased reflectance in the wavelengths of pigment emission compared to the samples with $SiO_2$ (Figures 5 and 6). The peak at 1609 $cm^{-1}$, present for all samples with FP and nanoparticles, can be assigned to the C=O stretching vibrations [56]. In addition, the area of OH vibrations, between 3200 and 3500 $cm^{-1}$ had an absorbance of weak intensity but was more pronounced for the samples with $TiO_2$ nanoparticles, especially unaged ones. Finally, the peaks at 1109 and 1345 $cm^{-1}$ are characteristic due to their changes after the ageing process. The peak at 1109 $cm^{-1}$ can be assigned to the plane bending vibration of the C–H mode due to the protonation [57]. The absorbance of this peak was decreased after the artificial ageing. The peak at 1345 $cm^{-1}$ can be assigned to the C=O stretching [58,59], pointing to the higher surface polarity of the coatings with $TiO_2$ nanoparticles. Its absorbance was slightly decreased after ageing.

Due to the photocatalytic activity of $TiO_2$ nanoparticles [37,60], and the higher number of chemical bonds that have changed their absorbance after the ageing process, it can be concluded that $SiO_2$ does not cause a decrease in the reflectance of the FCs to such an extent as $TiO_2$ because there are fewer reactions and interactions with the prepared TB and FP.

*3.4. Roughness Measurement*

Surface roughness was measured on the paper substrate and on all printed coatings, unaged and aged. In order to determine the surface structure of coated paper, the $R_a$ roughness parameter was determined. $R_a$ of the printing substrate was measured in order to analyze the surface structure of the basic material. Arithmetic mean deviation of the profile ($R_a$) was 0.3766 µm (SD 0.042). The results were expected, due to the fact that the structure of the coating on the paper surface is porous, full of small irregularities detected in the form of micro roughness that amounts to less than 1 µm (visible in Figure 2b).

In the printing process, by the application of TB on the paper substrate, the roughness was increased ($R_a$, from 0.3766 to 0.906 µm). Obviously, the microstructure of the paper substrate and the structure of the TB containing the small spheres of polymer dispersed in an aqueous medium (plasticizer) caused an additional increase in surface roughness. As it is visible in Figure 2, the micro voids caused further distortion of the plane surface which affected the small and additional increase in surface roughness. The addition of FP in the TB increased the roughness, as well. $R_a$ was changed from 0.906 to 2.259 µm. Due to the fact that pigments are insoluble in binders in which they are dispersed and that their size is from a few microns up to 100 µm, as can be seen in Figure 3, those results were expected.

Figure 8 presents the results of $R_a$ parameter measured on printed FCs with the addition of different concentrations of $SiO_2$ and $TiO_2$ nanoparticles. One can see the changes in surface roughness after 6 and 12 h of accelerated ageing.

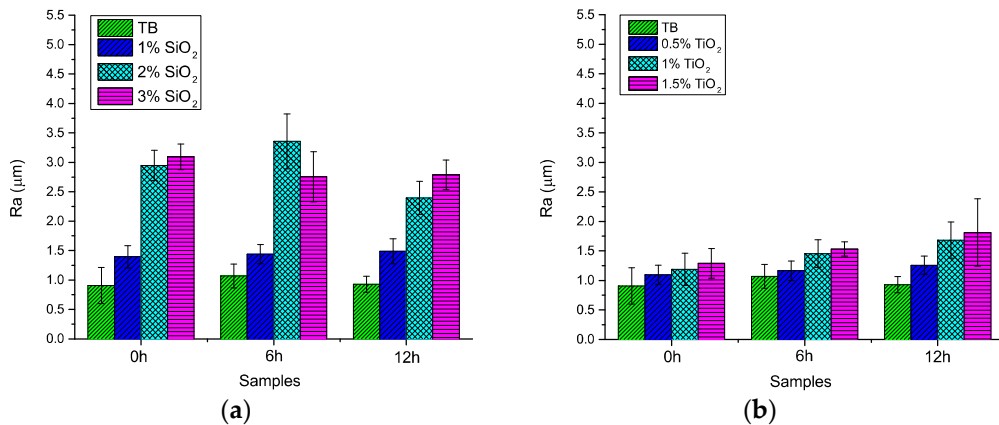

**Figure 8.** Results of the measurement of $R_a$ parameter of the unaged and aged FC with addition of (**a**) $SiO_2$ and (**b**) $TiO_2$ nanoparticles.

It is visible that FCs containing particles of $SiO_2$ have higher values of roughness in comparison to coatings printed with the addition of $TiO_2$. During the ageing process of the samples with the addition of $SiO_2$ and $TiO_2$, certain changes in roughness were detected (Figure 8). TB without the addition of nanoparticles showed no significant change in roughness during ageing (Figure 8a). However, one can see that the addition of 1% of $SiO_2$ caused a slight increase in the roughness during the ageing process (from 1.397 μm measured on unaged samples, to 1.439 μm measured on samples aged for 6 h and 1.488 μm measured on samples aged 12 h). The addition of 2% $SiO_2$ caused a marked increase in roughness after 6 h of ageing (from 2.947 μm measured on unaged samples, to 3.356 μm measured on samples aged for 6 h) and a decrease to 2.395 μm after 12 h of ageing. Ageing of the samples with 3% $SiO_2$ led to a decrease in roughness after 6 h of ageing (from 3.094 μm measured on unaged samples, to 2.755 μm).

After 12 h of aging, the roughness has practically not changed and amounts to 2.786 μm. The addition of different concentrations of $TiO_2$ to the FC caused an increase in surface roughness. Figure 8b shows a slight increase of the $R_a$ parameter for all $TiO_2$ concentrations. During the ageing process, an increase was detected for all samples (for 0.5% of $TiO_2$ from 1.096 μm measured on unaged samples, to 1.165 μm measured on samples aged for 6 h and 1.256 μm measured on samples aged 12 h). Coating consisting of 1% $TiO_2$ had an $R_a$ parameter of 1.189 μm measured on unaged samples, 1.454 μm measured on samples aged for 6 h, and 1.682 μm measured on samples aged 12 h. The results of roughness measured on coating consisting of 1.5% $TiO_2$ had $R_a$ parameters measured on unaged samples of 1.288, 1.533 μm on samples aged for 6 h, and 1.812 μm measured on samples aged for 12 h.

By observing Figure 8, it can be concluded that the roughness of the samples' surfaces increased with the increased concentration of the nanoparticles in the base. Furthermore, the roughness of the coating surfaces with added $SiO_2$ nanoparticles decreased after the ageing on the samples with higher concentrations of $SiO_2$ (2% and 3%). This could be assigned to the degradation process involving the breakage of C–Cl bonds in plastisol, visible at 837 cm$^{-1}$ in ATR-FTIR spectra (Figure 7b in Section 3.3), and to the other changes in the surface structure including other chemical groups detected by ATR-FTIR analysis. On the other hand, roughness of the coating surfaces with $TiO_2$ nanoparticles noticeably increased after ageing on the samples with higher concentrations of $TiO_2$ (1% and 1.5%). This could be related to the agglomerates of plastisol formed on the surface of the coatings that are visible in SEM images (Figure 14, Section 3.7).

### 3.5. Contact Angle of Water

Results of the contact angle measurements on the unaged and aged samples of the FCs with added nanoparticles are presented in Figure 9. They were obtained in order to

detect the changes of the surface polarity after the addition of the nanoparticles and after the artificial ageing.

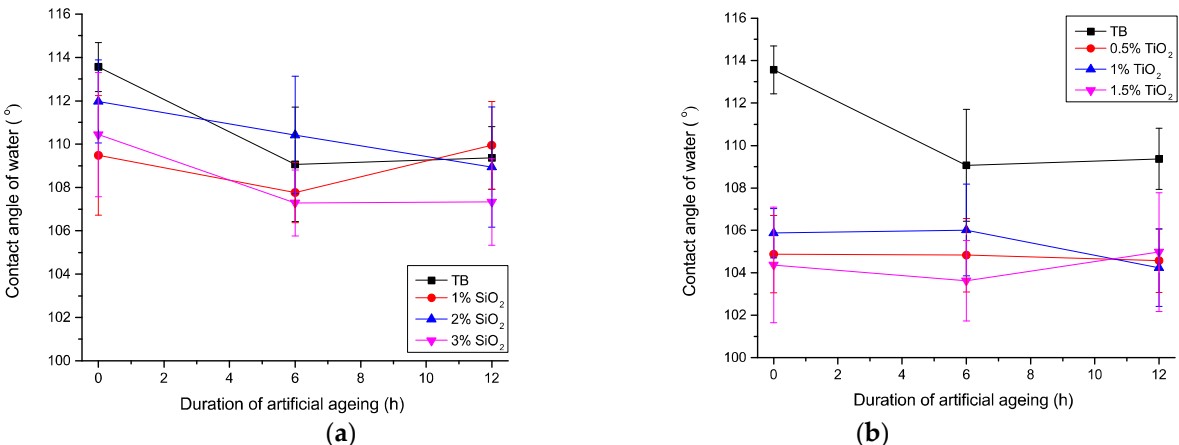

**Figure 9.** Contact angles of water on the unaged and aged FC with addition of (**a**) $SiO_2$; (**b**) $TiO_2$ nanoparticles.

In Figure 9, it is visible that the deviations in the water contact angle are fairly high. This can be assigned to the variable roughness as a result of the principle of screen printing technique; and to the heterogeneity of the printed coatings [61]. Nevertheless, it can be concluded that the surfaces of the coatings with added $TiO_2$ nanoparticles displayed significantly lower contact angles (Figure 9b) than the TB and the surfaces of the coatings with added $SiO_2$ nanoparticles (Figure 9a). Specifically, the maximal contact angle was measured on the unaged TB surface (113.56°). Minimal average contact angles were measured after the ageing process, when using the highest concentrations of the nanoparticles (107.3° on the coating with 3% of $SiO_2$, and 103.6° on the coating with 1.5% of $TiO_2$).

It can be concluded that the surfaces of the coatings with $TiO_2$ nanoparticles are more polar, possibly since more OH and C–O groups were detected in ATR-FTIR analysis compared to the coatings with $SiO_2$ (Figure 7). Furthermore, since SEM images (Section 3.7) revealed more protrusions and micro voids on the surfaces of the coatings with added $SiO_2$ nanoparticles. Higher values of contact angles on them could be related to the air traps due to their surface topography [62]. Since the roughness of the samples with 2% and 3% $SiO_2$ decreased after the ageing process, the contact angle of water decreased, as well.

In conclusion, polarity of the coating surface can be important in the cases of printing the additional motive (for example, bar code) beneath or on the top of the FC. Changes in the polarity of the coating surface could affect the adhesion between the printed layers [63].

*3.6. Bending Stiffness of Printed Samples*

Results of the bending stiffness measurements are presented in Figure 10. Considering the possible applications of FCs, bending stiffness is of importance when assessing the usability and durability of printed products.

Bending stiffness of the paper used in this research, without any print, was $4.5 \pm 0.25$ mNm. Observing Figure 10, one can conclude that the addition of the nanoparticles to the coating, especially of $SiO_2$, significantly improved the bending stiffness. This result is in accordance with previous research, where the incorporated $SiO_2$ nanoparticles improved various mechanical properties of the materials [64–66]. Specifically, the bending stiffness of the unaged coating with 1% of $SiO_2$ was $8.25 \pm 0.14$ mNm. After adding 3% of $SiO_2$ to the coating, bending stiffness was $11.31 \pm 0.69$ mNm, presenting the increase of 40%. $TiO_2$ nanoparticles did not improve the bending stiffness of the coating significantly. Maximal bending stiffness of the coating was achieved on the printed sample with 1.5% $TiO_2$, after 6 h of artificial ageing ($6.45 \pm 0.31$ mNm).

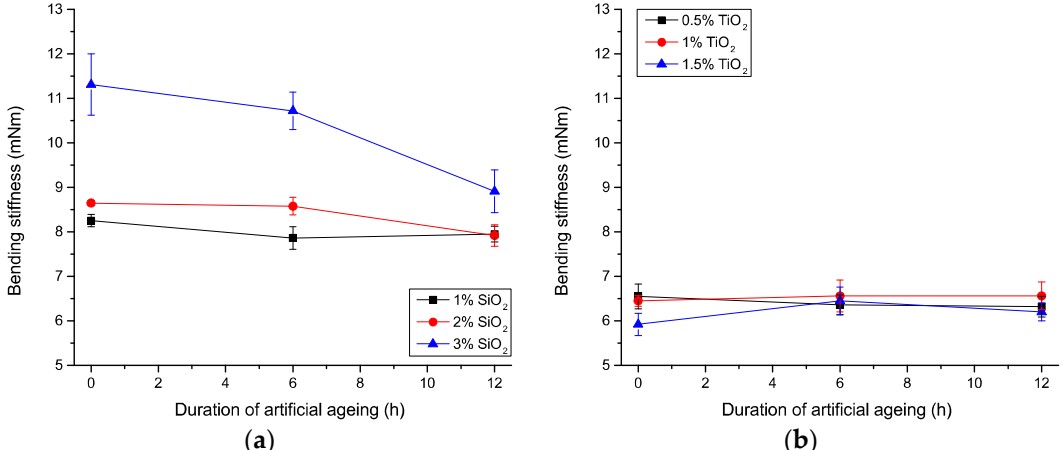

**Figure 10.** Bending stiffness of the unaged and aged FC with addition of (**a**) $SiO_2$; (**b**) $TiO_2$ nanoparticles.

The process of artificial ageing caused the changes of the bending stiffness for all samples: the higher the concentration of $SiO_2$ nanoparticles, the higher the decrease of the bending stiffness of the aged samples (Figure 10a). In contrast, the ageing process has caused the slight increase of the bending stiffness for the coatings with higher concentration of $TiO_2$ nanoparticles (Figure 10b). The decrease of the bending stiffness in general can be related to the start of the degradation of the coating with $SiO_2$ nanoparticles, while the increased bending stiffness after the ageing of the coatings with $TiO_2$ nanoparticles can be explained by the formation of the agglomerates of plastisol due to the interaction with $TiO_2$ observed in SEM analysis.

### 3.7. Microscopy and SEM Images of Printed Surfaces

Microscopic images of the chosen coating surfaces, taken at a magnification of $20\times$, are presented in Figures 11 and 12. All images were taken under UV light source to ensure the visibility of the fluorescence effect.

Observing Figures 11 and 12, one can visually confirm the results of the spectral reflectance measurements (Figures 4–6). The ageing process had a noticeable effect on the fluorescence of all samples. It is important to notice that the addition of $TiO_2$ nanoparticles resulted in a more saturated visual fluorescence effect on unaged samples than the addition of $SiO_2$. This phenomenon can be related to the higher spectral reflectance of the samples with $TiO_2$ (Figure 6).

The effect of the artificial ageing, i.e., the fading of the FC under a UV light source, is more pronounced on the samples with $TiO_2$ nanoparticles (Figure 12b,d) than on the samples with $SiO_2$ nanoparticles (Figure 11b,d). This is also in accordance with the spectral reflectance of the coatings and with the photocatalytic properties of nano-$TiO_2$.

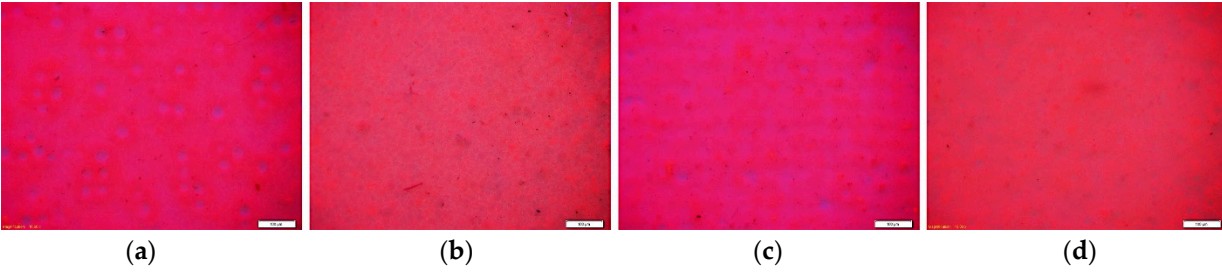

**Figure 11.** Microscopic images of coatings illuminated by UV light source (**a**) FC + 1% $SiO_2$—unaged sample; (**b**) FC + 1% $SiO_2$—aged for 12 h; (**c**) FC + 3% $SiO_2$—unaged sample; (**d**) FC + 3% $SiO_2$—aged for 12 h.

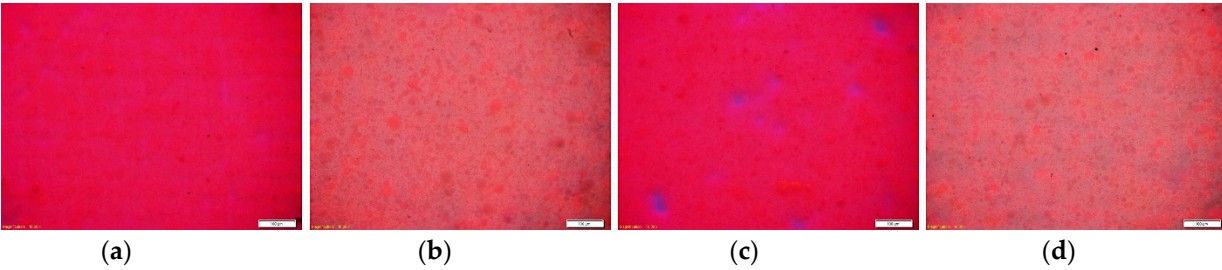

(a)  (b)  (c)  (d)

**Figure 12.** Microscopic images of coatings illuminated by UV light source (**a**) FC + 0.5% TiO$_2$—unaged sample; (**b**) FC + 0.5% TiO$_2$—aged for 12 h; (**c**) FC + 1.5% TiO$_2$—unaged sample; (**d**) FC + 1.5% TiO$_2$—aged for 12 h.

SEM images of the chosen unaged and aged samples are presented in Figures 13 and 14. As can be seen from the captured images, the surface of the coatings with TiO$_2$ is less irregular than in the case of the addition of SiO$_2$, although micro voids, polymer spheres, and some rare protrusions are still present (Figure 13). The most significant change is observed in the sample with 1.5% TiO$_2$ after 12 h of ageing (Figure 14). It seems that the polymer spheres from the TB (plastisol) tend to agglomerate among themselves into larger agglomerates after ageing. The reason for this could be the higher mass concentration of TiO$_2$, but additional analysis for further discussion is required.

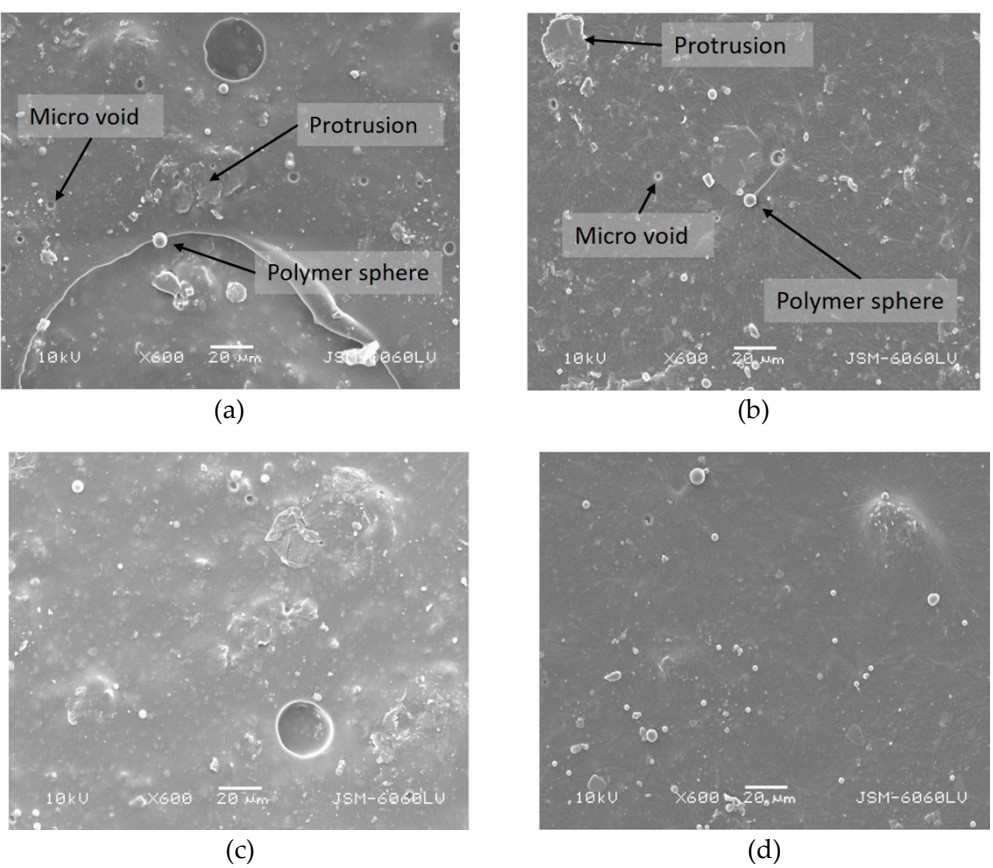

(a)  (b)

(c)  (d)

**Figure 13.** Surface morphology of samples (**a**) FC + 1% SiO$_2$—unaged, (**b**) FC + 0.5% TiO$_2$—unaged, (**c**) FC + 1% SiO$_2$—aged for 12 h, (**d**) FC + 0.5% TiO$_2$—aged for 12 h (mag. 600×).

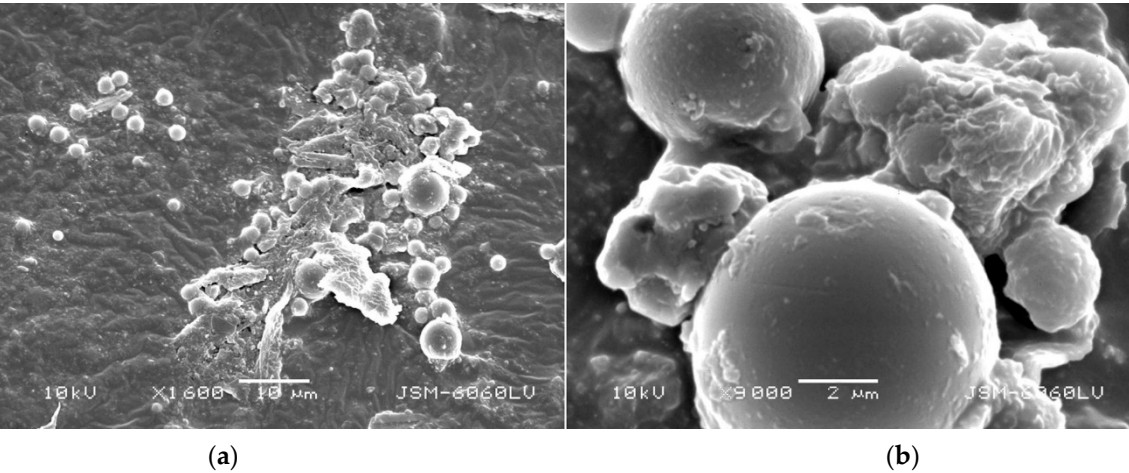

(**a**)                                                                                                                    (**b**)

**Figure 14.** (**a**) Surface morphology of sample FC + 1.5% TiO$_2$ aged for 12 h (mag. 1600×); (**b**) agglomeration of polymer spheres (mag. 9000×).

### 4. Conclusions

In this research, the properties of UV visible (daylight invisible) fluorescent coating modified by the addition of SiO$_2$ and TiO$_2$ nanoparticles was studied. Nanoparticles were added to the coating in different mass concentrations in order to evaluate the spectral reflectance and other properties of the produced coatings. UV fluorescent coatings with added nanoparticles were transferred onto silk-coated paper by the screen printing technique. Structural, surface, and mechanical properties and changes in coatings caused by an accelerated ageing process were analyzed, as well.

The results showed that the addition of nanoparticles caused different changes in unaged and aged printed coatings. Based on the experimental results, the following conclusions can be drawn:

- The addition of lower concentrations of SiO$_2$ (1%) proved more effective in the ageing protection of FC compared to the coatings with added TiO$_2$ nanoparticles that have photocatalytic properties.
- The addition of TiO$_2$ to the coatings enables an increased percentage (%) of spectral reflection and better visual effect of the printed coatings compared to other coating mixtures.
- The results of the coatings' surface structure analysis showed that roughness was increased with the increased concentration of the nanoparticles. By the addition of SiO$_2$, roughness was decreased after the aging process due to the degradation process. Roughness of the coatings with TiO$_2$ nanoparticles was increased after ageing on the samples with higher concentrations of TiO$_2$ due to the agglomerates of plastisol formed on the surface of the coatings, which is visible in SEM images.
- Surface analysis of coatings showed that TiO$_2$ caused a large increase in the polarity of the surface coatings in comparison to SiO$_2$. Due to this, FCs with the addition of TiO$_2$ could be successfully used for printing on water-based primers and inks. On the other hand, coatings with the addition of SiO$_2$ nanoparticles could be used for printing on non-polar primers and inks.
- The results of the bending stiffness showed that the addition of the nanoparticles to the coating, especially of SiO$_2$, significantly improved the bending stiffness of unaged samples (40%).
- According to the results presented in this research, one can conclude that the addition of nanoparticles can improve the structural and mechanical properties and the visual effect of FCs. With this in mind, the optimization of the nanoparticles' quantities will help with the functionalities of coatings, i.e., the ageing process, and at the same time could significantly increase the bending stiffness.

**Author Contributions:** Conceptualization, S.M.P. and T.T.; methodology, S.M.P. and T.T.; investigation, S.M.P., T.T., M.L. and U.S.E.; resources, S.M.P., T.T., M.L. and U.S.E.; writing—original draft preparation, S.M.P. and T.T.; writing—review and editing, S.M.P., T.T., M.L. and U.S.E.; visualization, S.M.P., T.T., M.L. and U.S.E.; funding acquisition, S.M.P. and T.T. All authors have read and agreed to the published version of the manuscript.

**Funding:** This research was funded by short-term scientific support provided by University of Zagreb.

**Institutional Review Board Statement:** Not applicable.

**Conflicts of Interest:** The authors declare no conflict of interest.

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
