# Peer review of "Effect of SiO2 and TiO2 Nanoparticles on the Performance of UV Visible Fluorescent Coatings"

_coatings, doi:10.3390/coatings11080928_

Round 1
Reviewer 1 Report
The authors investigated the effect of SiO_2 and TiO_2 nanoparticles on polymer coatings containing fluorescent chromophores. Several properties were measured on freshly prepared samples and artificially aged samples, i.e., samples exposed to light irradiation for 6 or 12 hours. The investigated parameters include the optical reflectance spectra, FTIR spectra, surface roughness, contact angle of water, bending stiffness, and fluorescence intensity as visible under a microscope. The main result is that the addition of a small amount of SiO_2 particles reduces the degradation of the fluorescence intensity due to aging, whereas TiO_2 particles have the opposite effect.
The results may be helpful for scientists and engineers working in the field of fluorescent coatings for technical applications. Hence, the data should be published. In my opinion, however, the paper can be shortened without compromising its content. Specifically, the discussion of the roughness parameter R_z can be omitted, since it contains the same information as R_a. Also, many of the SEM micrographs presented in section 3.7 can be omitted, since the same information is presented in section 3.4 - roughness measurement.
A specific remark: The term "UV fluorescent ink" is misleading, since it leads to the assumption that the chromophores fluoresce in the UV spectral region rather than in the red (Fig. 4).
Finally, the authors should try and improve the quality of the English.
Author Response
Dear Reviewer,
please see the attachment.
Thank you in advance.

Reviewer 2 Report
The Paper “ Effect of SiO2 and TiO2 nanoparticles on the performance of 2 fluorescent UV - responsive coatings” by Sanja Mahović Poljaček et al. describes the phenomenon of fluorescence response of the unaged and aged UV fluorescent coatings modified by addition of SiO2 and TiO2 nanoparticles.
The work presents an in-depth morphological characterization however for a work that proposes to study fluorescent coatings in my opinion the fluorescence measurements of the films are necessary.
Authors provide a Spectral reflectance analysis but from these data it is difficult to understand the contribute due to reflectance and due to fluorescence. An in depth analysis of fluorescence variation as intensity and position depending on the type and quantity of nanoparticles, and depending on the Aging and subsequent comparison of the effect on fluorescence properties due to different nanoparticles is necessary.
For this reason I can only consider the work after these analysis will be added.
Below some other suggestions
Reading the abstract and the introduction it is not clear that the work is about coatings on paper, it can only be seen in the "Materials and Methods" section. I suggest reporting and discussing the reasons for this choice in the abstract and introduction.
I suggest to clearly add also in the abstract the main results concerning the fluorescent UV responsive coatings. Example: In line with what is reported in the title does the addition of nanoparticles have a positive effect on the fluorescence intensity of the film?
In my opinion, the data on fluorescent films are scarce.
What intensity of emission do the films have before adding the nanoparticles? Does the addition of nanoparticles have an effect on the fluorescence intensity of the films? Are there any scattering effects? Is there evidence of aggregation quenching of the dye?
In Fig. 5 Authors compared spectral reflectance of unaged and aged coatings of TB_UVFP_1%SiO2and TB_UVFP_2%SiO2 and reported that concentrations of SiO2 (1%) results with the smallest decrease of the pigment’s reflectance after 12h of ageing. How do they perform the comparison? Please Discuss.
In "Characterization Methods" Authors reported the use of a Deuterium-Tungsten Halogen UV light source DH-2000 that has a generically Wavelength Range between ~215-2500 nm. Do they select a wavelength? I suggest to use only the wavelength excitation of the Red Pigment ( I suppose around 360-380 nm for an Europium dye) to partially or completely eliminate the optical brighteners contribute
The author often refer to UV fluorescent inks this means that the ink are fluorescent in the UV region between 200-400 nm. Is it correct?.
Pag 2 Line 45 “ fluorescence inks”, the correct form is fluorescent inks.
Pag 3 line 115 “TiO2 in concentrations higher than 1% can diminish the optical properties of coating” what kind of optical properties? Absorption, emission, scattering , transparency….please discuss.
Fig. 7 In the FTIR-ATR spectra Analysis of UV fluorescent coatings I suggest adding and discussing the FTIR-ATR spectra of the fluorescent pigment dispersed in the transparent base before and after aging.
Author Response

(The authors gave the same response as above.)

Reviewer 3 Report
This is an interesting study on the modification of fluorescent coatings. The aim is to enhance the fluorescence response. As the aim relates to objectives that are shared by numerous researchers in industry and academia alike, the work may be of value to others, and thus it can be published. The quality of the work can be significantly improved by several changes. Please address all of the following comments:
- Authors should be very clear about the type of coatings they are modifying, The results are relevant for similar coatings. For example the title should specify the type of coating and type of pigment being modified. I could not find any information about the so-called UVFP, which is stated to be UVFP -red by Cestisa.
- It is absolutely imperative that the authors specify the composition of the fluorescent pigment being modified, as the results relate entirely to this pigment. - what is the composition of this UVFP???
- What other UVFP are available on the market??
- How would we expect the TiO2 and SiO2 modifications to affect other types of UVFP?
- What is the rationale for choosing the pigments and the modification methods?
- How would the results differ if other phases of TiO2 were used?, e.g. anatase vs. rutile or mixtures of these phases, this is also true for SiO2
- How does particle size affect performance? Is it possible or worthwhile to introduce finer particle sizes in the coatings?
- What are the real-world implication of the findings reported in this work?
Author Response

(The authors gave the same response as above.)

Round 2
Reviewer 2 Report
Considering also the opinion of the other reviewers I can accept the manuscript even if the absence of the fluorescence measurements decrease the quality of the manuscript. In fact, reflectance measurements do not provide the same type of information and fluorescence measurements are required for any real future application
Reviewer 3 Report
The paper has been revised and is good enough for an MDPI journal.